# Mechanism of Iron Ion Homeostasis in Intestinal Immunity and Gut Microbiota Remodeling

**DOI:** 10.3390/ijms25020727

**Published:** 2024-01-05

**Authors:** Honghong Bao, Yi Wang, Hanlin Xiong, Yaoyao Xia, Zhifu Cui, Lingbin Liu

**Affiliations:** College of Animal Science and Technology, Southwest University, Chongqing 400715, China; bao519119@163.com (H.B.); hddszxf@163.com (Y.W.); xhl122598@163.com (H.X.); yaoyaoxia2018@126.com (Y.X.)

**Keywords:** iron, iron homeostasis, intestinal immunity, gut, gut microbiota

## Abstract

Iron is a vital trace element that plays an important role in humans and other organisms. It plays an active role in the growth, development, and reproduction of bacteria, such as *Bifidobacteria*. Iron deficiency or excess can negatively affect bacterial hosts. Studies have reported a major role of iron in the human intestine, which is necessary for maintaining body homeostasis and intestinal barrier function. Organisms can maintain their normal activities and regulate some cancer cells in the body by regulating iron excretion and iron-dependent ferroptosis. In addition, iron can modify the interaction between hosts and microorganisms by altering their growth and virulence or by affecting the immune system of the host. Lactic acid bacteria such as *Lactobacillus acidophilus (L. acidophilus*), *Lactobacillus rhamnosus* (*L. rhamnosus*), and *Lactobacillus casei* (*L. casei*) were reported to increase trace elements, protect the host intestinal barrier, mitigate intestinal inflammation, and regulate immune function. This review article focuses on the two aspects of the iron and gut and generally summarizes the mechanistic role of iron ions in intestinal immunity and the remodeling of gut microbiota.

## 1. Introduction

Gut microbiota plays a major role in the initial formation and enhancement of the human immune system. As medical research progresses, the association between gut microbiota and intestinal immunity has become increasingly important due to its close relationship with human health and potential novel therapeutics. Research on gut microbiota dates back more than a century, with Nobel laureate in Medicine, Ilya Mechnikov, being the first to delve into this field. He observed that elderly individuals in Bulgaria had a fondness for consuming yogurt, and hypothesized that a healthy gut microbiota could help extend life [1]. Subsequently, in 2005, Robin Warren and Barry Marshall were awarded the Nobel Prize in Physiology or Medicine for their discovery that *Helicobacter pylori* is the cause of human gastritis, stomach ulcers, and duodenal ulcers [2]. This exemplifies that the answer to many human health issues may not be solely within the host but could be related to the host microbiome. Recent studies have shown that gut microbiota can stimulate anti-tumor immune response by regulating CD8^+^ T cells, T helper 1 (Th1) cells, and tumor-associated bone marrow cells in order to improve tumor immune evasion [3,4]. Therefore, gut microbiota reprogramming is an essential scientific research approach for treating cancer and regulating the homeostasis of the tumor microenvironment. Iron ions are the key factors in sustaining homeostasis and play a vital role in maintaining the internal equilibrium of the body. Studies have shown that iron influences the homeostasis of gut microbiota, thereby showing a close correlation between them. However, any disturbance or remodeling of gut microbiota can cause an imbalance in the physiological function of the human body or other host environments, leading to health problems [5,6,7]. However, there are several different views on the definition of gut microbiota dysbiosis [8]. Researchers have different views on the disruption of internal environmental equilibrium. For instance, some have suggested three types of microecological imbalances, such as the “overgrowth of pathogenic organisms” and the “loss of symbiosis or diversity”, whereas others have classified them into four types: loss of key groups, reduction in diversity, changes in metabolic capacity, or proliferation of pathogens. This gives a broad range of definitions for the disruption of internal environmental balance [7]. As the microbiota dysbiosis of different diseases varies, it is important to investigate the true causes of their dysbiosis in order to develop effective therapeutic strategies to gain a better understanding of this field. This review article aims to provide a comprehensive overview of iron ion homeostasis, intestinal immunity, gut microbiota remodeling, and the correlations among them, thus laying a foundation for further exploration of the field.

## 2. Overview and Role of Iron

Iron is an essential trace element that is required by almost all organisms [9]. Organisms have evolved mechanisms to preserve and recycle iron internally. The average amount of iron in a grown adult is approximately 3–4 g, while the amount of iron lost daily is merely 1–2 mg [10]. In order to maintain iron balance, a healthy person must consume an equivalent amount of iron in their diet, mostly in the form of vegetables and meat. After the iron is absorbed into the body of an organism, it directly participates in the iron circulation mechanism. There are four main cell types that regulate iron circulation in the body (Figure 1): intestinal cells in the duodenum obtain iron from food; red blood cell precursors in the bone marrow add iron to the hemoglobin [11]; macrophages in the liver, spleen, and bone marrow reclaim iron; and liver cells store iron [12]. Iron circulation is regulated by several factors that can sense iron levels and, subsequently, regulate the expression of genes required for iron homeostasis, where hepcidin—ferroportin (FPN) interaction is a major regulatory mechanism. FPN is a transmembrane protein that mediates the circulation of iron, permitting it to be transported from the cells to the bloodstream [13]. A study showed a high concentration of FPN in the intestinal cells, macrophages, and liver cells [14]. Further, a study by Adriana Donovan et al. reported the importance of FPN in iron output and revealed that the absence of iron absence led to embryonic death [15], thereby demonstrating its critical role in development.

Iron has the capacity to donate or accept electrons in both the cellular and extracellular environments, which makes it a multifunctional catalyst in numerous enzymes involved in energy production, essential biosynthetic pathways, and the generation of reactive oxygen species (ROS) for host defense. Iron also coordinates with hemoglobin and myoglobin for oxygen transport and cellular storage, respectively [16]. Iron is present in living organisms in three forms: attached to protein side chains [17], complexed in the heme porphyrin ring [18], and as a part of iron-sulfur clusters [19]. When not in these controlled chemical environments, iron can disrupt cell and tissue functions. Iron, in its ferrous form, also plays an important role in the mechanism of iron death [20,21], so the uptake, storage, and use of iron need to be strictly controlled. In fact, all organisms have developed complex mechanisms to regulate iron levels [22,23,24].

## 3. Iron Homeostasis

### 3.1. The Transportation of Iron in and out of Cells

Iron homeostasis is a process that involves the uptake, storage, and utilization of iron and is managed by proteins, which control iron transport, isolation, and sensing. These proteins regulate iron levels at both cellular and biological levels based on the environmental concentration of iron, erythropoiesis requirements, body load of iron, and redox stimulation [25]. Iron homeostasis is determined by the activity of six cell types such as visceral endoderm and placental cells located outside the embryo, absorbent intestinal cells, erythroid precursors, recovered macrophages, and hepatocytes [26]. These cells are categorized into three main groups: intestinal epithelial cells, which transfer iron through the apical and lateral basement membrane (placental cells and intestinal cells); the cells that store and release iron when necessary (macrophages and hepatocytes); and the cells that use iron but do not release it until they die (erythroid precursors). The activities of these cells regulate the iron pool bound to transferrin [27]. Cells need to interact with iron; however, iron cannot pass through the cell membrane without assistance, which is provided by two types of transmembrane transport proteins, including Iron-Ion Transporter SLC11A2 (DMT1, Nramp2, and DCT1) and iron transport proteins (FPN, SLC40A1, IREG1, and MTP1) [28]. SLC11A2 acts as an iron importer, which takes up iron into the cells, whereas iron transporters act as iron exporters, which transfer the iron out of the cells [29]. Notably, FPN is the only known mammalian cell iron transporter protein. SLC11A2 is present in the apical membrane of the proximal intestinal epithelial cells, while FPN is found on the outside of the basement membrane. In addition, SLC11A2 is also present in the endosomes of erythroid precursors, where it facilitates the transferrin cycle for iron uptake [30]. On the other hand, FPN is not expressed in erythroid cells but is distributed in the placenta, intestine, reticuloendothelial macrophages, and hepatocytes. The presence of these transmembrane transporters in the above-mentioned six cell types is essential for maintaining iron balance, known as iron homeostasis in the body.

In addition, iron intake by the body from the diet mainly includes heme iron and non-heme iron (Fe^3+^). After absorbing into the intestinal epithelial cells, the non-heme iron is reduced to Fe^2+^ by cytochrome b at the brush edge of the duodenal villi and then transported by divalent metal-ion transporter 1 (DMT1) [31,32]. Studies have demonstrated that duodenal cytochrome b (Dcytb) is a homolog of the cytochrome b561 family and located at the fringes of intestinal epithelial cells. Dcytb is equipped with a resistance ascorbic acid binding site and uses ascorbic acid as an electron donor to reduce Fe^3+^ to Fe^2+^; this reaction is mediated by an ascorbyl acid-dependent reductase [33]. Red meat is the primary source of heme iron, which is a combination of cerebral globin, myoglobin, and porphyrins in the blood. Heme iron is less affected by other factors and is released by the absorption of Haem carrier protein (HCP-1) and the action of heme oxygenase. It is then absorbed by the intestinal cells, red blood cells, liver cells, etc., and further enters the iron cycle, thereby maintaining iron homeostasis in the body [34,35]. The transport process of iron inside and outside the cell is shown in Figure 2.

### 3.2. Mechanisms Exist to Maintain Iron Equilibrium in the Body

Iron homeostasis is regulated by the hepcidin—ferroportin axis, which controls the absorption of iron from dietary sources, macrophages, and body stores, as well as its systemic distribution [36]. Hepcidin can induce the degradation of iron transporters, thus controlling the entry of iron into the bloodstream [37,38]. Hepcidin has been found to be able to prevent toxicity brought about by too much iron. Studies have shown that hepcidin is able to prevent toxicity caused by excessive iron through two mechanisms. Firstly, when the concentration of hepcidin is high, it can block the outward conformation of iron transporters, thus preventing the efflux of iron from cells. Secondly, hepcidin can induce the endocytosis and degradation of iron transporters, leading to their permanent removal from the cell surface. This second mechanism is expected to occur at lower concentrations of ferritin; however, if the concentration of hepcidin subsequently decreases, it will still have long-term effects [38].

At the moment, it has been established that iron metabolism is mainly regulated by DMT1. This is distinct from membrane iron transporter 1 (FPN). DMT1 is the main iron transporter protein (divalent metal-ion transporter 1) and assists in the uptake of non-heme iron in the majority of cells [39]. The expression of DMT1 is regulated by the iron concentration through various translation and degradation pathways to maintain iron homeostasis. The iron responsive element-iron regulatory protein (IRE/IRP system) is essential for the management of iron metabolism in cells, overseeing the translation of iron transporters and correspondingly adjusting intracellular iron levels. The mRNA of the iron transporter protein contains 5′ IRE, which under iron deficiency conditions, binds to the iron regulatory proteins IRP1/2 and inhibits the translation of iron transporters, thus restricting iron output and preserving cellular iron. However, in duodenal cells, the absence of 5′ IRE in the iron transporter transcripts reduces the cells’ response to their own iron levels, making them more sensitive to the iron demand from plasma ferritin concentration [39]. During iron deficiency and anemia, low levels of ferritin cause intestinal cells to continuously release iron, activating the HIF system (especially HIF2α). This activation further increases the transcription of iron transporters, as well as other mRNA-encoding proteins involved in dietary iron uptake (Dcytb and DMT1). This coordinated increase in top and basal lateral iron transport leads to an overall increase in duodenal iron uptake [40].

## 4. Role of Iron Homeostasis and the Harmful Effects of Its Disturbance

### 4.1. Effect of Iron Homeostasis on the Human Body

Iron ions are the most crucial metal ions in the body and are mainly present in the form of Fe^2+^ and Fe^3+^, which are interchanged by the gain or loss of electrons. This gives iron its redox ability, allowing it to play a role in the production of hemoglobin, myoglobin, cytochrome oxidase, etc., as well as activating xanthine oxidase and other activities in the body [41,42]. However, the redox potential of iron ions can lead to the production of free radicals, which damage intracellular biomolecules. In order to ensure an optimal iron level, the body relies on iron transporters, such as the iron intake, storage, and release proteins, as well as iron regulatory proteins, which collaborate to maintain iron homeostasis. The imbalance of iron metabolism-related proteins in the body leads to iron metabolism disorders, which can be divided into two categories: iron deficiency diseases, such as iron deficiency anemia [43], which causes fatigue and hair loss; and iron overload diseases, such as hereditary hemochromatosis [44]. Recent studies have also linked systemic or local iron metabolic disorders to the development of chronic diseases, such as chronic anemia [45], renal anemia [46], cardiovascular disease, neurodegenerative disease [47], alcoholic liver disease [48], etc. Therefore, restoring iron homeostasis in the body is essential for the treatment of these diseases.

### 4.2. Role of Iron Homeostasis in the Treatment of Cancer

Recently, researchers have focused on ferroptosis due to its potential in the treatment of lung, liver, and breast cancers. This iron-dependent form of programmed cell death is distinct from apoptosis, necrosis, and autophagy [49,50]. Ferroptosis involves the catalysis of highly expressed unsaturated fatty acids on the cell membrane, leading to lipid peroxidation and subsequent cancer cell death [51]. Additionally, ferroptosis was reported to regulate the antioxidant system (glutathione system) to reduce the levels of the core enzyme glutathione peroxidase 4 (GPX4) [52]. GPX4 is a key enzyme, regulating glutathione (GSH) levels, and a major factor in ferroptosis. Ferroptosis is a non-apoptotic form of cell death, which is caused by the accumulation of iron ions within the cell, leading to an increase in toxic ROS production. Due to this regulatory pathway, numerous researchers are attempting to increase oxidative stress in the cancer cells in order to disrupt the redox balance and induce ferroptosis [53]. Currently, numerous studies have attempted to explore how ferroptosis can be employed to enhance the treatment of tumors and cancers [54,55,56]. For instance, some researchers used nanomaterials to introduce iron ions into cells, while others added reducing agents to activate a reduction in iron ions to ferrous ions, thereby inducing the Fenton response [57].

### 4.3. Consequences of Impaired Iron Homeostasis

Iron plays several biological roles, such as oxygen transport, ATP production, energy metabolism, and DNA synthesis and repair; therefore, it helps to promote the proper functioning of the cells and body [58]. Impaired iron homeostasis will lead to senescence or cell death and lead to a variety of diseases, such as Parkinson’s disease, neurodegenerative diseases, and mental disorders. In the body, iron deficiency or physiologically low levels of iron will lead to the occurrence of cellular inflammation. Alessia Pagani et al. [59] demonstrated that a decrease in hepcidin levels due to iron deficiency could increase the levels of inflammatory factors in mouse stem cells. However, hepcidin injection reduced the lipopolysaccharide (LPS)-induced increase in inflammatory cytokines in iron-deficient (ID) mice. Lin et al. [60] showed that iron was essential for both microorganisms and hosts, and its deficiency could reduce the body’s bactericidal ability, as well as increase the risk of serious infections. Additionally, excess iron concentration can cause oxidative stress, which can disrupt epithelial tight junctions and intestinal barrier permeability, thereby leading to increased inflammation and lipid peroxidation. Li et al. [61] demonstrated that both the lack and excess of dietary iron had detrimental effects on intestinal morphology, inflammation, and function in pigs. The imbalance of iron homeostasis (low or high iron levels) causes these issues, which leads to various diseases.

## 5. Regulation Mode of Iron Homeostasis

Iron homeostasis involves careful coordination among intestinal iron absorption, the uptake and release of cellular iron, and iron storage. Due to the absence of a physiological mechanism for iron excretion in humans, control of intestinal iron absorption is the main mechanism, ensuring the overall iron balance. Most of the iron required by the body is acquired by the intestinal cells from the diet; a fraction of the absorbed iron is transferred to the bloodstream [62], whereas the remaining iron in the intestine is expelled from the body through the intestinal lumen. The intestinal iron absorption is mainly regulated by the action of hepcidin, a circulating 25-aa hormone produced in the liver and filtered into the urine [63]. Hepcidin negatively regulates cellular iron release, thereby limiting the iron present in the plasma [64]. As a negative regulator, hepcidin also affects iron homeostasis by interacting with the only known receptor cellular iron exporters, FPN, thereby decreasing intestinal iron absorption and iron release from macrophages and increasing plasma iron levels, which leads to a greater iron load. This interaction also results in endocytosis and lysosomal breakdown, thereby preventing iron from entering the plasma, causing hyposideraemia, and restricting erythropoiesis [63,65]. Clinically, there are numerous “regulators” of the plasm hepcidin concentration. The increase in iron storage or inflammation increases the level of hepcidin, which can be decreased by hypoxia, increased erythropoiesis activity, or testosterone. In cases of conflicting effects on hepcidin expression, such as non-transfusion-dependent thalassemia [66], negative regulation of erythropoietic activity is the most effective regulatory mechanism. Hepcidin expression is physiologically regulated at the transcriptional level by a complex network formed by the interaction of multiple proteins and cascades. Although the molecular mechanism regulating the expression of hepcidin is not fully understood yet, progress has been made in the investigation of human diseases or mouse models of hemoglobin dysregulation [67].

## 6. Effects of Iron Homeostasis on the Intestine

### 6.1. Effects of Iron Homeostasis on Intestinal Epithelial Cells

Cells can take up iron through the transferrin pathway; however, most cells absorb it through receptor-mediated endocytosis of diferric transferrin. A study demonstrated that immature erythroid cells mainly relied on TfR1-mediated endocytosis of ferritin [68]. Furthermore, dietary iron is absorbed from the intestinal lumen by proximal small intestinal cells through transferrin-independent pathways. This absorption process is generally divided into three stages: from the intestinal lumen to the apical membrane, cellular transport, and then into the blood circulation via the lateral basement membrane, during which iron binds to apolipoprotein transferrin [25,69].

The intestinal cells take up iron from the intestinal lumen and transport it into the blood circulation through the apical divalent metal transporter-1 (DMT1), which transports Fe^2+^ from the lumen into the cell. On the other hand, iron is transported from the intestinal cell into the blood circulation through the basolateral transporter, called the ferrotransporter [70]. Intestinal epithelium, a major component of the intestinal barrier, is mainly composed of absorptive intestinal epithelial cells, goblet cells, enteroendocrine cells, and tufted cells [71]. After absorption by intestinal epithelial cells, iron binds to “putative chaperones” and is transported internally. For example, PCBP2 can transfer ferrous iron from DMT1 to the appropriate intracellular sites or ferroportin and could function as an iron chaperone [72]. If the cellular iron level is high, it is stored in the iron storage protein ferritin and eventually lost due to cell shedding. On the other hand, if the cellular iron level is low, it is transported through the outer basement membrane into the blood circulation to bind to the transferrin [73]. Iron transporters regulate intestinal iron homeostasis, while transferrin receptor 1 is expressed on the outer basement membrane of the intestinal epithelium. Previous studies have shown that ferritin can pass through the intestinal epithelial cells and accumulate in the blood [74].

### 6.2. Effects of Iron Homeostasis on the Remodeling of Gut Microbiota

Gut microbiota are incredibly diverse in terms of both microbial and genetic composition, and the composition of species numbers depends on their location in the gastrointestinal tract [75]. For instance, the stomach contains around 10^1^ microbial cells per gram, while the duodenum, jejunum, ileum, and colon have 10^3^, 10^4^, 10^7^, and 10^12^ microbial cells per gram, respectively [76]. This shows an increasing number of microbial cells from the proximal to the distal part of the digestive system. Notably, the large intestine is home to over 70% of all microorganisms in the body, which can greatly impact the host’s health. Additionally, bacterial diversity is higher in the lumen and lower in the mucus layer [77].

The intestine has a high bacterial abundance, having a diverse range of biochemical and metabolic activities, which interact with the host’s physiology. These bacteria aid in breaking down the indigestible polysaccharides and producing vitamins essential for health and are also involved in the development and differentiation of the intestinal epithelium and the host’s immune system [78].

Human gut microbiota are largely composed of two anaerobic bacterial phyla: Bacillota and Bacteroidota. Other phyla, such as Pseudomonadota, Verrucomicrobiota, Actinobactota, Fusobacteriota, and Cyanobacteria, are present in small proportions [79]. Studies have suggested that an imbalance in the Bacillota to Bacteroidota ratio might increase disease susceptibility; the mice with induced aging exhibited a lower Bacteroidota to Bacteroidota ratio [80]. Additionally, the lower abundance of phylum Proteobacterium, which is associated with a higher abundance of genera *Bacteroides*, *Prevotella*, and *Ruminococcaceae*, is linked with healthy gut microbiota. Therefore, it is important to maintain a balanced gut microbiota for a symbiotic relationship with the host.

### 6.3. Role of Gut Microbiota

The colonization of an infant’s gut by microbiota begins instantly after birth. By the time the infant reaches 3 years of age, the diversity and composition of gut microbiota are comparable to adults, and they remain relatively stable throughout adulthood, which is essential for the gut’s well-being [81]. In 1991, Lynn Margulis proposed a symbiotic relationship between gut microbiota and host, which was later widely referred to as Holobiont [82]. Subsequent research on Holobint has focused on restoring altered microbiota to prevent and treat diseases. Van de Guchte et al. [83] noted that the disruption of homeostasis can lead to a transition from a healthy to a pre-disease and disease state, which may explain why there has been an increase in inflammatory diseases and obesity in the Western world. Other studies further extended this concept, arguing that the unity of the host symbiont is the true organism, and free-living species of a single entity are merely abstract concepts [84]. On the contrary, regarding intestinal immunity, the microbial colonization of the gut begins in infants immediately after birth. At about 3 years of age, gut microbiota reach a composition and diversity similar to adults and remains more or less stable over time in adulthood [85]. Gut microbiota triggers myeloid cells, such as macrophages and dendritic cells (DCs), in the gut, thereby triggering innate and adaptive immunity, as well as inflammatory responses [86]. Additionally, intestinal epithelial cells, which act as the first line of defense against pathogens, are important for activating the immune system and protecting the host. For gut microbiota to remain dominant, they must be able to adapt to the gut environment and fend off the invasion and colonization of foreign pathogens in both direct and indirect ways [87]. In conclusion, the gut and its microbiota are interdependent, and the presence of beneficial microbiota is essential for gut health and the overall health of the organism. As shown in Figure 3, we enumerate the effects of different gut microbes and their metabolites on human organs and cell sites, as well as the main functions of the various metabolites.

### 6.4. Regulatory Effects of Iron Homeostasis on Gut Microbiota Remodeling

Currently, the mechanism through which alterations in the iron levels influence the structure and activity of gut microbiota is still unknown. Nevertheless, previous studies reported that iron has a significant effect on gut microbiota environment. Tompkins et al. [88] conducted a study to evaluate the impact of varying levels of dietary iron on gut microbiota composition of mice. They observed that the total number of anaerobic bacteria in the colon of mice with iron deficiency and those with iron overload were comparatively low, suggesting that both iron deficiency and trivalent iron supplementation could inhibit the growth of bacterial microorganisms, which they believe may be due to redox environments. Another study by EA Mevissen-Verhage et al. [89] demonstrated that the infants given a milk preparation with iron supplementation had lower levels of *bifidobacteria* but higher levels of *Bacteroidota* and *Escherichia coli* (*E. coli*) compared to those taking milk preparations without iron supplementation. Similarly, Zimmermann, Michael B. et al. [90] showed that iron fortification/iron supplementation could increase the abundance of *Enterobacteriaceae* and decrease *Lactobacillus*. They suggested that the increase in the abundance of Enterobacteriaceae might be mainly due to an increase in commensal and non-pathogenic *E. coli*. Moreover, Cheng et al. [91] conducted a comprehensive analysis of gut microbiota using 16S rRNA gene sequencing and showed that iron deficiency anemia (IDA) could significantly reduce the abundance of Bacillota and increase Bacteroidota, Pseudomonadota, and Patescibacteria. Similarly, Balamurugan et al. [92] discovered that iron-deficient women had lower levels of *Lactobacillus* in their feces compared to normal women. These studies suggest that insufficient iron can reduce the abundance of commensal beneficial bacteria, such as *Lactobacillus*, while the excess of iron can increase the abundance of harmful bacteria, such as Bacteroides and *E. coli*. It is essential to maintain a balanced iron level for preserving intestinal and overall health. Additionally, it can be inferred from these examples that iron deficiency or excess can influence gut microbiota and microbiota; however, the exact mechanism of its effect is still unknown and requires further investigation.

## 7. Effect of Gut Microbiota on the Body and Intestinal Immunity Levels

The human microbiome and host have grown together and become a single entity known as the holobiont [93]. In a balanced microecosystem, microbiota stimulates the body’s immune response when there is an invasion of foreign pathogenic bacteria and performs other important functions as well. However, the dysbiosis of gut microbiota causes an imbalance of the intestinal microecosystem, leading to a disruption of the body’s functions and a decrease in the body’s immune response, thereby making it more susceptible to diseases [94,95]. A study demonstrated that imbalances in the gut microbiota of mice can result in a weakened immune system [96]. They had an increased abundance of aerobic bacteria and *Clostridium* and a decreased abundance of beneficial bacteria, such as *Bifidobacteria*, *Lactobacillus*, and butyrate-producing bacteria [97,98]. *Bifidobacterium* species are usually present in healthy human intestines, and changes in their proportion and composition are common in these diseases [81]. These changes can trigger the body to activate both specific and non-specific immunity responses, thereby strengthening the body’s own protective abilities and preserving the internal balance of gut microbiota.

### 7.1. Regulatory Effects of Gut Microbiota on Intestinal Immune Microenvironment Remodeling and the Human Body

The human gut has evolved distinct regional immune characteristics, which are maintained by a mature intestinal mucosal immune system [99]. This system is separate from the innate and adaptive immune systems, which are highly differentiated systems present in specific intestinal regions. Moreover, the intestinal lymphoid tissues, consisting of isolated lymphoid follicles, Peyer’s patches, mesenteric lymph nodes, and intestinal epithelial cells, are associated with this complex system [100]. In contrast to the central and peripheral immune organs, the special intestinal structure, function, and microenvironment can induce innate and adaptive immune responses, which form the local immunity of the gut [101]. This local immunity provides some protection, even if the first and second defense barriers are breached. Additionally, gut microbiota plays a crucial role in this environment. Their composition is linked to the maintenance of homeostasis in the gut. Various microorganisms, such as *Bacteroides* and *Helicobacter pylori,* colonize the gut environment to help the host resist pathogenic factors, prevent obesity, and assist with digestion. For instance, gnotobiotic mouse transplants showed that transferring the gut microbiota of ob/ob mice with leptin deficiency or diet-induced obesity to sterile mice showed a greater tendency for obesity compared to wild type littermates or mice given a healthy low-calorie diet, suggesting that gut microbiota can lead to obesity in mice, which are overweight or have a disrupted leptin regulation [102]. In addition, Johnston et al. [103] reported a reduced intestinal inflammation in mir-21-deficient mice during acute colitis and miR-21-deficient mice with altered microbiota. This provides protection against the development of colitis. *Ruminococcus gnavus* exacerbated inflammatory bowel disease (IBD) by promoting a tolerogenic immune response by producing capsular polysaccharides [104]. A study by Zhang et al. [105] showed that the intestinal colonization of *H. pylori* could prevent chronic experimental colitis by modulating the Th17/Treg balance and transforming macrophages into an anti-inflammatory M2 phenotype. These findings indicated that the intestinal immune microenvironment might be affected by changes in gut microbiota, leading to the development of various gut microbiota- and microenvironment-related diseases. Using the interaction between beneficial bacteria and the intestinal tract, it is possible to reduce the symptoms of intestinal disease, including diarrhea and gastrointestinal infections [106].

### 7.2. Regulatory Effects of Gut Microbiota on Autoimmune Diseases

Autoimmune diseases are chronic inflammatory diseases, which cause extensive damage to multiple organs and systems due to the excessive activation of the body’s autoimmune system, as well as the production of autoantibodies [107]. Numerous studies have shown that gut microbiota can produce specific molecules, which can induce the production of inflammatory cells and factors to regulate the integrity of the intestinal mucosal barrier and refine the multiple functions of mucosal immunity [108,109]. A recent study [110] showed that the presence or absence of gut microbiota affected the occurrence of autoimmune stress in mice. The study also found that programmed cell death protein 1 (PD-1) receptor-deficient mice exhibited autoimmune disease, which could be prevented by removing intestinal bacteria from these mice. The PD-1 receptor can also suppress the immune system, and its absence causes the overactivation of the immune system, which, in combination with gut microbiota, can trigger several immunosuppressive diseases, such as autoimmune encephalomyelitis [111] and human immunodeficiency virus (HIV) infection [112]. In systemic lupus erythematosus (SLE) patients, the causative commensal bacterium *Enterococcus quail* could translocate from the small intestine to the liver site, probably due to driving the expression of interferon-related genes and autoantibody production. The early colonization of gut microbiota can directly shape the B-cell pool through the symbiotic bacteria-induced expression of the ribonucleoprotein Ro60, which triggers a cross-immune response, thereby causing autoimmune disease in susceptible individuals [113]. Recently, Wang et al. [114] demonstrated that modulating the structure of gut microbiota using prednisone might be an effective way to achieve the therapeutic effects of glucocorticoids (GCs) for the treatment of SLE while avoiding side effects. They also highlighted the role of gut microbiota in regulating immune responses, which might be beneficial for the treatment of autoimmune diseases. Lee et al. [115] determined that gut microbiota consist of microorganisms that can direct the pro-inflammatory and anti-inflammatory immune responses of the central nervous system. Consequently, it is imperative to comprehend the part of the microbiota in guiding the host immune response for the purpose of devising suitable treatments for multiple sclerosis.

### 7.3. Gut Microbiota Can Shape Innate and Adaptive Immunity for Immune Homeostasis

The signaling of gut microbiota is crucial for the development of the immune system. The manipulation of gut microbiota, either using antibiotic therapy or microbiota reconstitution, provides key evidence for their role in immune homeostasis. Gut microbiota can regulate the local intestinal immune system and exert certain effects on the systemic immune response [116]. The intestinal antigen-presenting cells (APCs) have evolved to protect the body from infection while still allowing for immune tolerance of commensal gut microbiota [117,118]. DCs in Peyer’s patches produce high levels of interleukin-10 (IL-10) compared to those produced by splenic DCs activated under similar conditions [119]. Intestinal macrophages are located near gut microbiota and have a special phenotype called “inflammatory incompetence”, which means that they cannot produce pro-inflammatory cytokines in response to microbial triggers, such as toll-like receptor (TLR) ligands [120]. Similar to those in other tissues, these macrophages are highly phagocytic and bactericidal. However, phagocytosis does not usually cause significant inflammation in mice and humans [121,122,123,124]. Macrophages are key players in maintaining intestinal homeostasis, as well as key sentinels of the intestinal immune system.

Neutrophils are an important component of the innate immune system. Gut microbiota can systemically regulate neutrophils [125]. Being neutropenic is a particularly striking phenotype of germ-free animal (GF) rats. In addition, GF rats exhibited impaired superoxide anion and nitric oxide production, as well as decreased phagocytosis in their peripheral blood neutrophils. Interestingly, the translocation of GF rats back to a conventional or specific pathogen-free (SPF) environment did not restore the normal superoxide anion phenotype [126]. A recent mechanistic study showed that cytoplasmic receptor-nucleotide oligomerization structural domain 1 (NOD1) could enhance the killing activity of bone marrow neutrophils by recognizing peptidoglycan from gut microbiota [127]. Wen et al. [128] discovered that the interplay between intestinal microorganisms and the innate immune system is a major epigenetic factor in altering the vulnerability to T1D (type I diabetes), implying that there is a connection between intestinal microorganisms and innate immunity, which can further modify the status of immune diseases by altering the composition of intestinal microorganisms. These studies demonstrate the role of microbiota in enhancing neutrophil function.

### 7.4. Effect of Short-Chain Fatty Acids Produced by Gut Microbiota on Intestinal Immunity and Autoimmunity

Gut microbiota have a symbiotic relationship with their human hosts and are essential for intestinal health and stability. They produce various metabolites, among which short-chain fatty acids (SCFAs) are among the most important and abundant ones [129,130]. SCFAs are produced by the fermentation of non-starch polysaccharides and resistant starch by anaerobic colonic bacteria [131]. SCFAs have powerful anti-inflammatory effects and can modulate the immune system, GPCRs such as GPR41, GPR43, and GPR109A on the surface of epithelial and immune cells bind to SCFAs, and their transportation or diffusion into host cells can result in their metabolism and/or inhibition of histone deacetylase (HDAC) activity. SCFAs have a variety of functions, including enhancing the epithelial barrier, encouraging immune tolerance, and sustaining intestinal homeostasis through particular mechanisms [115], thereby providing protection against numerous diseases, such as cardiovascular disease and IBD [132,133]. For instance, butyrate can strengthen the epithelial barrier and induce antimicrobial peptides [134], such as regenerated islet-derived protein 3 (Reg3) γ and β-defensins [135]. It can also reduce inflammatory responses and maintain intestinal immune homeostasis, thereby potentially providing protection against IBD [136]. In addition, propionate could provide long-term radioprotection, attenuate hematopoietic and gastrointestinal syndromes, and reduce pro-inflammatory responses in mice [137]. Actually, the specific role of SCFAs in gut microbiota and homeostasis is not clear; in some cases, more butyrate is better, but in others, more propionate or acetate are better, so we need further studies to disentangle the roles of SCFAs.

### 7.5. Effect of Intestinal Probiotics on Intestinal Immunity

Currently, researchers are focusing more on probiotics to promote human intestinal health. Beneficial microorganisms, such as probiotics, can regulate the composition of gut microbiota and enhance immunity [138] by keeping the epithelial barrier intact, preventing the attachment of pathogens to the intestinal surface, and regulating and maturing the immune system [139]. Furthermore, probiotics can effectively treat certain diseases by influencing gut microbiota [140,141,142]. Evidence has been presented that probiotics, gut microbiota, and immunity are closely linked [143]. Future studies on the mechanism of probiotics in modulating gut microbiota and improving immunity should be conducted, which might be beneficial for improving the overall health and quality of human life.

## 8. Effects of Gut Microbiota Composition on the Host

### 8.1. Composition of Gut Microbiota Can Affect the Immune and Internal Physiological Environment of the Host

Changes in the composition of gut microbiota can cause intestinal issues and alter gut microbiota composition, which can affect the regulation of the host’s physiological activities, including the development and functioning of the immune system. Gut microbiota can affect the pathogenesis of human immunity and metabolism, such as innate and adaptive immune responses and metabolic processes [144]. The innate immune response can be impacted by inflammatory vesicles, cytokines, microglia, and TLRs, thereby increasing inflammation in the body. Adaptive immune responses can be affected by T cells and mast cells, thereby further aggravating the inflammatory state [145]. In addition, gut microbiota can further affect host physiology by influencing the metabolic activities of the organisms, such as bile acid metabolism, the trimethylamine N-oxide (TMAO) pathway, and fatty acid metabolism [146,147].

### 8.2. Gut Microbiota Can Regulate the Development of the Central Nervous System

Gut microbiota are essential for maintaining metabolic and immune health. Numerous studies have demonstrated that gut microbiota can affect the development of the central nervous system. Several studies have proven that the SCFAs affect the gut—brain axis through indirect means. This communication system comprises signals from the gut, which are transmitted to the brain’s central nervous system via the Vagus nerve, thus affecting some of the brain’s activities and controlling the emotions of the organism [148,149], as shown in Figure 4. Studies have shown that gut microbiota are correlated with diseases, such as depression [150], anxiety disorders [151], Parkinson’s disease [152], and others. *Bifidobacteria* in the gut can enhance the anti-tumor effects of PD-L1 inhibitors, as well as DC function and CD8 T cell-mediated anti-tumor mechanisms [153]. These diseases facilitate the development of neurological lesions.

### 8.3. Homeostasis of the Intestinal Microenvironment

The intestinal microenvironmental homeostasis depends on multiple factors, including host genetics, intestinal immune system, gut microbiota and metabolites, and intestinal barrier integrity and function [110,154,155,156]. In terms of host defense against intestinal pathogens, the mucosa-associated immune system protects the gut at an early stage through the dynamic function and coordinated cellular interactions of epithelial cells, DCs, and macrophages to recognize foreign pathogens. Moreover, gut commensal bacteria are essential for the maturation of the innate immune system. The interactions between DCs and natural killer cells can initiate the intestinal immune response by activating DCs and macrophages through different pathogen-associated molecular patterns [157,158]. The intestinal immune system, gut microbiota, and mucosal barrier are crucial for maintaining the continual equilibrium of the intestinal microenvironment. Moreover, gut microbiota have a major role in sustaining and regulating dynamic homeostasis. Probiotics form a significant part of gut microbiota and can boost the potency of gut microbiota to fight off invading enteric pathogens through competitive rejection processes and the production of bacteriostatic and bactericidal substances. Furthermore, they are specifically beneficial for sustaining and triggering the mucosal immune response.

A study showed that avian-specific probiotics, including *Lactobacillus acidophilus* and *Streptococcus faecalis*, could reduce the colonization of *Clostridium jejuni* in market-age broilers [159]. Probiotics were reported to competitively suppress pathogens that attack the intestinal mucosa and enhance the integrity of the intestinal epithelial barrier by activating gastrointestinal innate or adaptive immune systems. Studies have also shown that *Lactobacillus* and *Bifidobacterium* can eliminate *Chlamydomonas* jejuni from immunologically active and immunodeficient mice [160]. Probiotics interact with intestinal epithelial cells to promote mucosal immunity by initiating a pro- or anti-inflammatory response, which contributes to the regulation of intestinal homeostasis. Prebiotics and non-digestible food substances selectively promote the growth of probiotics and human health through nutrient enrichment and the modulation of gut microbiota and the immune system. Postbiotics have been shown to improve gut health by strengthening the gut barrier, reducing inflammation, and promoting antimicrobial activity against gut pathogens [133,161].

## 9. Conclusions

As is shown in Figure 5, this review describes the correlations among iron homeostasis, gut microbiota and the microenvironment, and intestinal immunity. It has been found that iron can not only maintain the balance of the internal environment to avoid diseases, such as anemia and hereditary hemochromatosis but also affect the composition and abundance of gut microbiota. Modifying the composition of gut microbiota can further impact gut immunity and the development of host diseases, as well as regulate the development of the central nervous system. All these three components collaborate to preserve the homeostasis of the microenvironment, thus keeping the stability of the entire host life system. Moreover, iron ions can also be used to induce ferroptosis to treat and destroy cancer cells. Jiang et al. [162] investigated the drug resistance model of the 4T1 mouse triple-negative breast cancer (TNBC) cell line and determined that TYRO3 has a significant correlation to the resistance of cancer patients to anti-PD-1/PD-L1 treatments. This implies that TYRO3 can be used as a predictive biomarker for patient stratification, with the potential to enhance treatment results. A study conducted by Chao Mao demonstrated that the lncRNA P53RRA could directly interact with the functional domain of G3BP1, resulting in abnormal accumulation of p53 in the nucleus to induce cell arrest and ferroptosis and then inhibit lung cancer progression [163].

## 10. Look Forward to the Future

At present, the advancement of medical technology has improved people’s health and quality of life, yet cancer treatment still mainly relies on traditional chemotherapy, which can cause significant harm to the body. To address this urgent need, drug-targeted therapy combined with iron-induced iron death could be a potential solution. Although research is still in its early stages, there is hope in the future as technology progresses that this new treatment method can be applied to cancer patients. Moreover, iron-induced changes in gut microbiota can regulate the immune system of the host, maintain intestinal homeostasis, and prevent inflammation, thereby becoming a potent novel approach for medical treatment. Therefore, this research approach demonstrates its practical value for our society in the future.

## Figures and Tables

**Figure 1 ijms-25-00727-f001:**
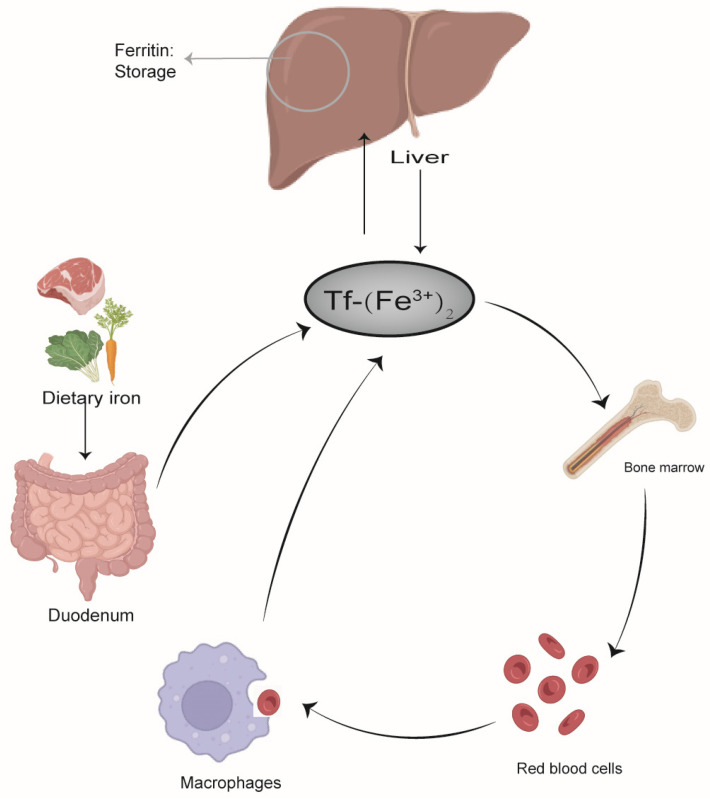
The four major cell types are responsible for controlling the iron cycle in the body. Dietary iron is absorbed by the small intestinal cells in the duodenum and binds to transferrin in plasma, which is mainly used to produce hemoglobin for new red blood cells. Macrophages recover most of the iron in the body by engulfing red blood cells and breaking down their heme fraction, and then reloading it onto lipotransferrin. Too much iron is stored in the liver’s ferritin. Tf, transferrin.

**Figure 2 ijms-25-00727-f002:**
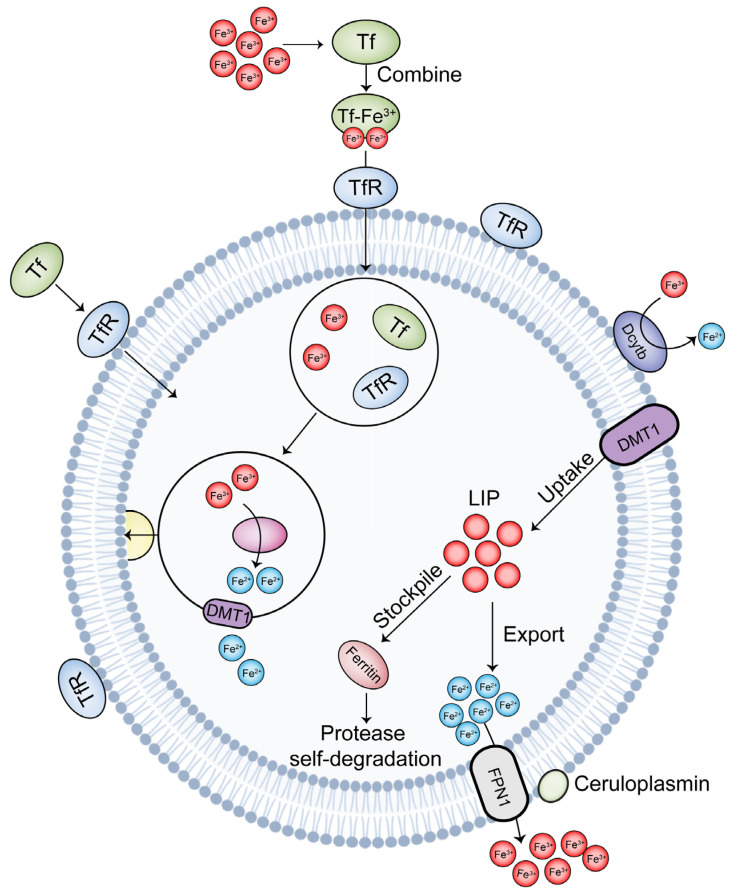
The transport of iron inside and outside cells. After binding to transferrin, iron ions are taken into the cell by protein receptors on the cell membrane and are separated from transferrin in an acidic environment. Enzymes reduce the ions to ferrous ions, and then they are excreted into the cell. Additionally, Dcytb can also convert iron ions to ferrous ions, and DMT1 mediates them into the cell. Some of the iron ions are stored as ferritin, whereas the rest are expelled from the cell, thus maintaining the iron ion content. Tf, transferrin; TfR, transferrin receptor; DMT1, divalent metal-ion transporter 1; FPN1, ferroportin 1; Dcytb, duodenal cytochrome b; LIP, iron pool.

**Figure 3 ijms-25-00727-f003:**
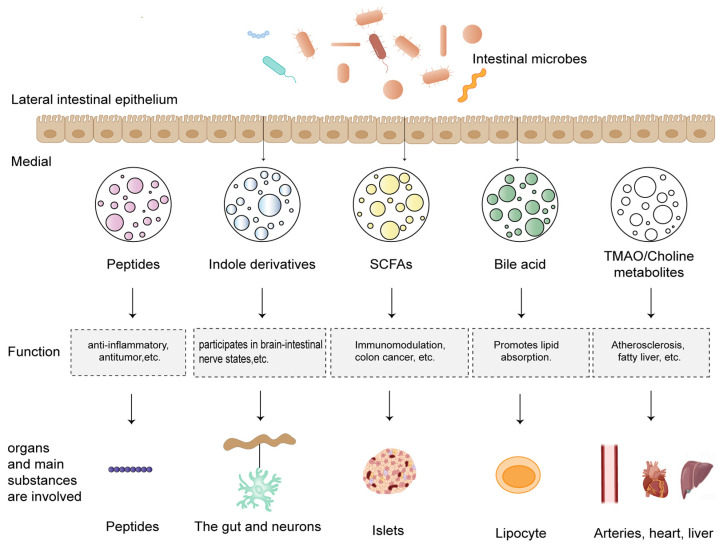
Examples of the functions of major metabolites of intestinal microbes. Gut microbiota are located on the outside of the intestinal epithelium, with the intestinal epithelial cells acting as a separator between them and the inner environment. Nevertheless, metabolites created by gut microbiota can penetrate channel proteins and enter the inner intestine, resulting in a variety of effects by going through different cell cycle pathways. This figure mainly gives examples of the roles of the most important metabolites and the organs or cells in which they act.

**Figure 4 ijms-25-00727-f004:**
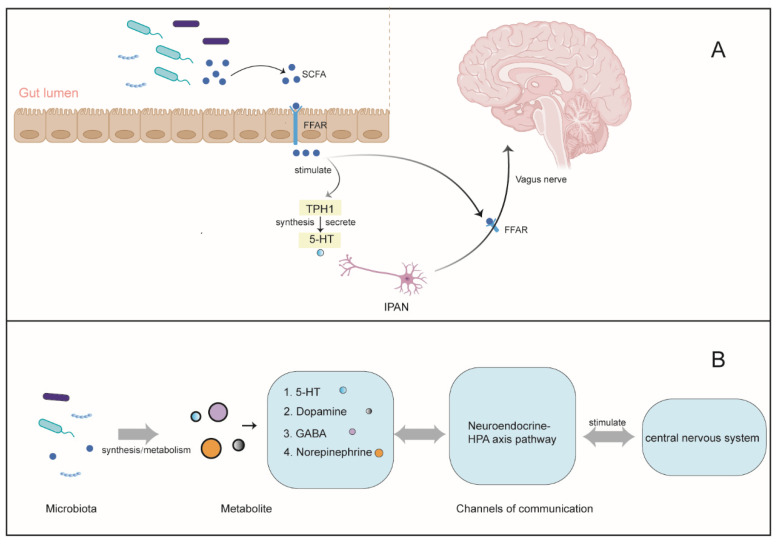
Microorganisms and the brain—gut axis. (**A**) Serotonin (5-HT) as a critical regulator of gut-brain-microbiome axis signaling. SCFAs can enter the bloodstream from the intestinal cavity via FFARs, which can then activate TPH1 to produce and release 5-HT in ECC cells. This free 5-HT can be transmitted to the Vagus nerve through IPAN, thus influencing the brain’s central nervous system. Additionally, SCFAs can bind fatty acid receptors on epithelial immune cells and nerve cells (e.g., the Vagus nerve), thereby regulating downstream processes such as movement, secretion, and enterocerebral signal transduction. Additionally, (**B**) other metabolites produced by the gut microbiota are involved in neural activity. Gut microbiota produce various metabolites, such as 5-HT, dopamine (a precursor of Neurohormone), GABA, Norepinephrine, etc., which can influence neural activity. These metabolites can be transported through the nerve Endocrine system, HPA axis pathway, intestinal mucosal barrier, and blood—brain barrier, altering sleep, anxiety, depression, and other brain emotional activities. SCFAs, short-chain fatty acids; FFARs, free fatty acid receptors; TPH1, tryptophan hydroxylase 1; 5-HT, serotonin; ECC, enterochromophilia; GABA, γ-Aminobutyric acid; IPAN, intrinsic primary afferent neuron.

**Figure 5 ijms-25-00727-f005:**
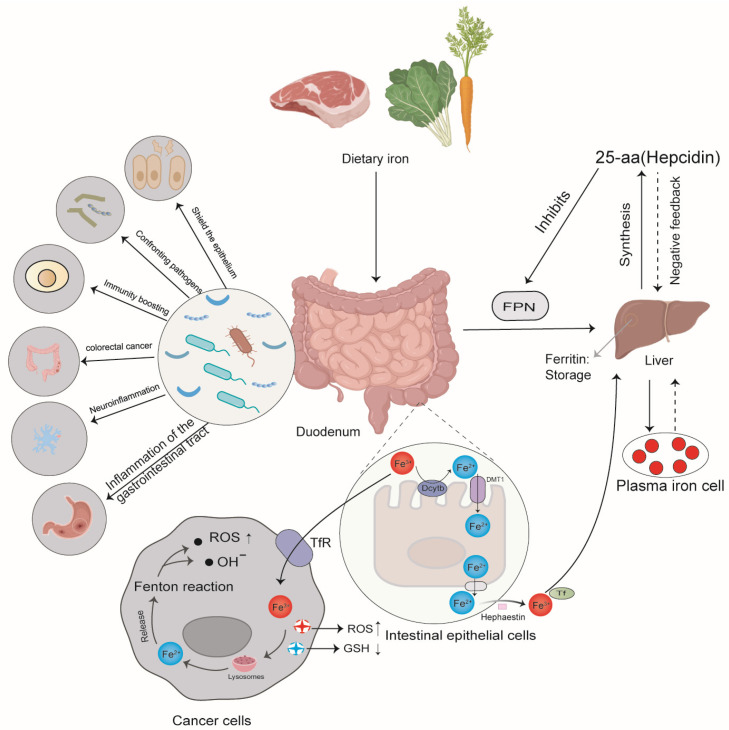
The key regulators of iron homeostasis and major iron cycling pathways. The human body obtains iron mainly through the ingestion of food, such as meat and vegetables. The iron is absorbed into the body through the intestinal epithelial cells, which are regulated by two major proteins: SLC11A2(FPN), an iron importer, and SLC40A1(DMT1), an iron exporter. These two proteins work together to maintain the balance of iron ion concentration in the intestinal cells. Additionally, the hepcidin hormone is able to inhibit the synthesis of ferroportin, thus regulating the transport of iron ions. This hormone is also regulated by a negative feedback loop, which reduces its effect and allows the transport of iron to continue in an orderly manner. The ferroptosis pathway is also involved in this cycle, which involves the entry of free trivalent iron into cancer cells via transferrin. This process generates a large amount of toxic hydroxyl radicals and reactive oxygen species (ROS) through the Fenton reaction, resulting in an increased level of oxidative stress that leads to programmed cancer cell death. The figure also reveals the influence of gut microbiota on human health. Gut microbiota can be advantageous, yet they can also cause illnesses, like abnormal gastrointestinal metabolism. FPN, ferroportin; Tf, transferrin; Dcytb, duodenal cytochrome b; DMT1, divalent metal-ion transporter 1; TfR, transferrin receptor; GSH, glutathione. ↓: downregulation; ↑: upregulation.

## Data Availability

Data sharing is not applicable to this article as no datasets were generated or analyzed during the current study.

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
