# Peer review of "Mechanism of Iron Ion Homeostasis in Intestinal Immunity and Gut Microbiota Remodeling"

_ijms, 2024, doi:10.3390/ijms25020727_

Round 1
Reviewer 1 Report
Comments and Suggestions for Authors
This manuscript deals with an exciting iron and gut microbiota metabolism and homeostasis topic. The authors had a great topic, but the manuscript's organization could be more precise. Authors need to emphasize the relevance of their work in the conclusion section. Also, they need to improve the discussion of the cited articles in the manuscript.
I've included my detailed comments and suggestions in the attached file.

Comments on the Quality of English LanguageExtensive editing of the English language is required. Some sentences could be more precise, and the whole manuscript has so many typos.
Reviewer 2 Report
Comments and Suggestions for Authors
This review article by Bao et al. presents a comprehensive overview of the role of iron ion homeostasis in intestinal immunity and gut microbiota remodeling, highlighting its significance in human health and disease treatment. The authors effectively delineate the complex interactions between iron metabolism, gut microbiota, and the immune system, offering insights into the therapeutic potential of modulating these interactions. I am quite open to looking at a revised version if the authors could address some major and minor issues in a satisfactory fashion, which I describe in more detail below.
Major issues:
1. Clarification of Iron Homeostasis Mechanisms: The manuscript successfully outlines the essential role of iron in various biological processes. However, it would benefit from a more detailed explanation of the molecular mechanisms governing iron homeostasis. This could include the regulation of iron transport proteins and their interaction with cellular pathways (Section 3)​​.
2. Impact of Iron on Gut Microbiota: The article discusses the effects of iron on the composition of gut microbiota (Section 6.4)​​. It would be beneficial to further explore how variations in iron levels, both deficiency and excess, specifically alter the gut microbiota's composition and functionality, contributing to health and disease states.
3. Additionally, I recommend enhancing the description of the relationships between gut microbes, metabolites, and host immunity in section 6 and 7. The current manuscript overlooks several comprehensive studies that have systematically explored these interactions. Notable examples include the works of Akshit Goyal et al. (Nature Communications 2021) and Jaeyun Sung et al. (Nature Communications 2017), which offer detailed insights into these relationships. To strengthen the paper, it would be beneficial for the authors to integrate and summarize these key studies more thoroughly.
4. Clinical Relevance and Application: While the review emphasizes the potential therapeutic applications of understanding iron-gut microbiota interactions, particularly in cancer treatment (Section 10)​​, it would be helpful to include more concrete examples or case studies demonstrating these applications. This would provide a clearer pathway from theoretical understanding to practical medical interventions.
Minor comments:
1. Expansion of the Introduction: The introduction provides a good overview but could be expanded to include a brief history of research in this field, thereby situating the current review within a broader context (Section 1)​​.
2. Line 30: “regulating homeostasis of tumor microenvironment” -> “regulating the homeostasis of tumor microenvironment”
3. Line 59: “capacity of donating or accepting electrons” -> “capacity to donate or accept electrons”
4. Line 105: “in blood” -> “in the blood”
5. Line 251: “In conclusion, gut” -> “In conclusion, the gut”
6. Line 267: “When iron level is low” -> “When the iron level is low”
7. Line 363: “germfree animal” -> “germ-free animal”
Comments on the Quality of English LanguageMinor editing of English language required.
Round 2
Reviewer 1 Report
Comments and Suggestions for Authors
The authors have addressed all my comments and suggestions well. I have one last question.
Line 353: The authors say Lepidoptera is a bacterial genus, but Lepidoptera is a bug. Do you think that phrase is accurate? please clarify
Comments on the Quality of English Language
All scientific names must be in italics. Minor editing of the English language is required
Reviewer 2 Report
Comments and Suggestions for Authors
The authors answered the questions that I have raised. I do not have further comments.
Author Response
Dear expert Reviewer:
we want to express our gratitude for your time and effort in reviewing this manuscript and for providing valuable editing suggestions. Thank you once again.
Best wishes!
Sincerely yours,
Authors